# Barriers to and Facilitators of Adherence to Clinical Practice Guidelines in the Middle East and North Africa Region: A Systematic Review

**DOI:** 10.3390/healthcare8040564

**Published:** 2020-12-15

**Authors:** Saja H. Almazrou, Sarah I. Alfaifi, Sumayyah H. Alfaifi, Lamees E. Hakami, Sinaa A. Al-Aqeel

**Affiliations:** 1Clinical Pharmacy Department, College of Pharmacy, King Saud University, Riyadh 14511, Saudi Arabia; salmazrou@ksu.edu.sa; 2College of Pharmacy, Jazan University, Jazan 45142, Saudi Arabia; 201501335@stu.jazanu.edu.sa (S.I.A.); 201500337@stu.jazanu.edu.sa (S.H.A.); 201501314@Stu.jazanu.edu.sa (L.E.H.)

**Keywords:** clinical practice guidelines, dissemination, implementation

## Abstract

The current review aims to investigate the barriers to and facilitators of the adherence to clinical practice guidelines (CPGs) in the Middle East and North Africa (MENA) region. English language studies published between January 2010 and May 2019 were searched on PubMed, Embase, and EBSCO. The barriers were categorized as clinician-related factors, such as lack of awareness of familiarity with the CPGs, and external factors, such as patients, guidelines, and environmental factors. The search identified 295 titles, out of which 15 were included. Environmental factors, specifically lack of time, resources, incentives, availability, and costs of treatment or diagnostic tests, training, and dissemination plans were the most commonly identified barriers. The familiarity with or awareness of healthcare professionals about the guideline, guideline characteristics, lack of agreement with the guidelines and preference in clinical judgment, physician self-efficacy, and motivation were reported to a lesser extent. Few studies reported on the compliance of facilitators with the guidelines including disseminating and advertising guideline materials, education and training on the guidelines, regulatory and financial incentives, and support from institutions. The review highlights that the studies on barriers to and facilitators of compliance with CPGs in the MENA region are limited in number and quality.

## 1. Introduction

Clinical practice guidelines (CPGs) pertain to recommendations on optimizing patient care based on an appraisal of the quality of relevant evidence, comparison of benefits and harms of alternative options, and value judgments on the importance of such benefits and harms [1]. Guidelines are intended to decrease inappropriate variations in clinical practice and improve the quality and safety of healthcare, aid in resource allocation, and provide standards for performance evaluation [1].

However, implementing guidelines, such as translating evidence from the guidelines into practice [2], is a challenging process for CPG users [3,4]. The median proportion of self-reported adherence to guideline recommendations is 36% (IQR 30–56%) [5]. Evidence suggests that a considerable variation is observed in the pattern of “leakage” in the utilization of clinical guidelines at different steps: awareness, agreement, adoption, and adherence. For instance, leakages were observed between adoption and adherence for drug recommendations and between awareness and agreement for medical management recommendations [5].

The success of guideline implementation is dependent on corresponding implementation strategies and barriers [3,4]. Previous systematic reviews investigated the barriers and facilitators that influence the adherence of clinicians to guidelines. Cabana et al. [6] were one of the first systematic reviews on factors that restrict complete adherence by physicians to the guidelines. The review identified 76 articles, where 293 potential barriers were categorized as clinician barriers (i.e., lack of familiarity, awareness, agreement, motivation, self-efficacy, and outcome expectancy) and external barriers (i.e., patient, guideline, and environmental factors). Francke et al. [7] carried out a search strategy in 2006 and identified 12 systematic reviews on factors that influence the implementation of CPGs. The authors then classified the factors into those related to the characteristics of the guidelines (e.g., awareness of the existence of the guideline and familiarity with content), implementation strategies, professionals, patients (e.g., co-morbidity), and the environment (e.g., lack of support, staff, and time).

In 2016, Fischer et al. [3] categorized the barriers to guideline implementation into physician knowledge and attitude (personal factors), guideline-related factors, and external factors. Moreover, Correa et al. [8] conducted a systematic metareview and identified barriers in the social-political context (i.e., absence of a leader, difficulties in teamwork, and lack of agreement with colleagues), health system (lack of time, financial problems, and lack of specialized personnel), guidelines (lack of clarity and credibility of evidence), health professional (lack of knowledge about the CPG and confidence in oneself where relevant), and patients (negative attitude toward implementation, lack of knowledge about the CPG, and sociocultural beliefs).

To the best of our knowledge, studies that reviewed these barriers and facilitators in the context of the Middle East and North Africa (MENA) region are lacking. Correa et al. [8] identified 25 systematic reviews published up to 2018 on the barriers to and facilitators of the implementation of CPGs. In the majority of the identified reviews, the primary studies were mainly conducted in North America or Europe. Three reviews included primary studies from countries in the MENA region and focused on venous thromboembolism [9], hypertension [10], and evidence-based medicine implementation [11]. However, variations in the socioeconomic context, healthcare policies, and financing between countries may lead to different barriers to guideline implementation. Therefore, the current review aims to identify the barriers to and facilitators of the compliance of healthcare professionals to CPGs in the MENA region. Identifying such factors will help infer recommendations for designing interventions for CPG implementation and future research in the region.

## 2. Materials and Methods

The review is reported in line with the Preferred Reporting Items for Systematic Reviews and Meta-Analyses (PRISMA) guidelines. The review protocol was registered in PROSPERO (CRD CRD42020151134).

### 2.1. Selection Criteria

To be eligible for inclusion, studies had to report healthcare providers’ perceived barriers and facilitators of adherence to CPGs in the MENA region. Studies that described the current CPG practice in the MENA region, extent of compliance of CPGs, or clinician attitude toward or knowledge about the guidelines without describing the barriers to and enablers of compliance and implementation were excluded. Primary studies that used qualitative, quantitative, and mixed methods were included in the review. Unpublished data, abstracts, and conference proceedings were excluded. The search was restricted to papers published in the English language between January 2010 and May 2019. No restriction was placed on setting (i.e., hospital or primary care setting) or type of guidelines (i.e., general or disease-specific and national or international).

### 2.2. Search Strategy

Three bibliographic databases, namely, PubMed, EMBASE through the Ovid interface, and EBSCO were searched. Appendix A provides the full search strategy for PubMed (Appendix A).

### 2.3. Data Extraction

Bibliographic records from each database were uploaded into the Mendeley Desktop 1.19.4 reference management software, and duplicates were removed. Study selection occurred in two stages. First, three reviewers (LE, SH, and SI) screened the titles and abstract and evaluated eligibility. Second, two independent reviewers (LEA, SHA, SIA, or SAA) examined the full text of potential studies. Disagreements were resolved by a third reviewer (SHA).

One reviewer (LEH, SHA, or SIA) extracted data using a data extraction form implemented in Microsoft Excel. One reviewer (SAA) reviewed the extracted data for accuracy and completeness, which included year, study objectives, study design, setting, participants, and sample size.

### 2.4. Quality Assessment

Two reviewers (LEH, SHA, SIA, or SAA) independently appraised each study, and disagreements were resolved through a discussion between them. The Critical Appraisal Skill Program (CASP) was applied for qualitative studies [12], whereas the Center for Evidence-based Management critical appraisal checklist for cross-sectional studies [13] was applied for relevant studies. Appropriate tools according to the method were used to assess the quality of studies using mixed methods to identify barriers and facilitators.

### 2.5. Data Synthesis

The barriers and facilitators described in the Results sections of the included studies were extracted and grouped using a taxonomy of barriers to and facilitators of compliance of clinicians to CPGs [6]. The framework categorized barriers into factors that influence physician knowledge, attitudes, or behavior. Factors related to clinicians’ knowledge include awareness of the CPG and familiarity with CPG recommendations. Moreover, factors related to clinicians’ attitudes include an agreement with CPG recommendations and belief that implementation may not lead to the desired outcome (outcome expectancy) or that clinicians cannot perform the recommendations (self-efficacy), and lack of motivation. The external factors that restrict clinicians’ compliance to CPG recommendations include patients, guidelines, and environmental factors. Previous reviews on barriers to and facilitators of CPG implementation have used this comprehensive framework [2,6].

## 3. Results

### 3.1. Literature Search

The search identified 295 titles. Figure 1 displays the search results and processes of screening and selecting studies for inclusion. Forty-seven full publications were examined, out of which 15 met the inclusion criteria.

### 3.2. Study Characteristics

The majority of studies (*n* = 9) were published after 2015 (Table 1). The authors of the papers were commonly from Saudi Arabia (*n* = 5) and Palestine (*n* = 4). Six and five studies were conducted in hospital and ambulatory settings, respectively. In addition, seven, five, two, and one studies involved physicians, physicians and nurses, decision-makers, and nurses, respectively. The guidelines investigated were related to diabetes mellitus (*n* = 3), asthma (*n* = 2), obstructive pulmonary diseases (*n* = 1), antimicrobial prophylaxis (*n* = 1), smoking cessation (*n* = 1), pediatric patients with severe sepsis (*n* = 1), and others that were not disease-specific (*n* = 4). Lastly, nine, four, and two studies used the quantitative approach, quantitative approach, and mixed methods, respectively.

### 3.3. Study Quality Assessment

Wahabi and Alziedan [14] used mixed-methods and a chart review to identify the extent of compliance with guidelines. Then, the authors conducted interviews to identify barriers to adherence of the guideline recommendations. Therefore, quality assessment was focused on the qualitative portion of the study. Elsadig and Scott [15] invited prescribers in cardiology in Sudan for interviews to explore their views about the use of guidelines in clinical practice. A survey was then conducted among the doctors in the hospitals to examine the views of a larger population of prescribers. The CASP tool was used to assess the interview part, whereas the abovementioned checklist was used to assess the survey.

Table 2 presents the results of the quality assessment of the qualitative studies. The purposive snowballing sampling technique was used in four studies [16,17,18,19], whereas two studies targeted clinicians in specific hospitals [14,15]. Although the cited studies adequately described the selection of participants, no information was provided on the recruitment process, such as the original number invited and those who declined. The included studies described the methods of data collection adequately. For instance, two studies presented the topic guides as Appendix A [14,19], one adequately described the topic guide content [16], one described the method used to build the topic guide [15], and four discussed the saturation of data [16,17,18,19]. Analysis was conducted using a thematic framework [17,18,19], grounded theory analysis [16], and qualitative content analysis [14], whereas one study did not report the method used for analysis. The main concern regarding the quality of qualitative studies was that although rigorous, data analysis specifically failed to address the role of the researchers and point out potential bias during the formulation of research questions and data collection including sample recruitment and the rationale for deriving themes from the data.

All quantitative studies used the cross-sectional questionnaire methodology with sample size ranging from 44 to 401. Three studies [20,21,22] discussed the content and face validity of the questionnaires, whereas two studies [21,22] addressed construct validity and internal consistency. Only one study discussed the framework for questionnaire development [22]. Four studies reported that the instruments were based on previous questionnaires used in published literature without reporting any information on the validity of the original questionnaires [20,23,24,25]. Moreover, three studies reported sample size calculation and targeted all eligible participants [21,22,24]. The major concern in terms of quality was the lack of reporting on the method of the participant selection process to enable judgment on whether selection bias occurred and the representativeness of the sample (Table 3).

### 3.4. Barriers to and Facilitators of CPG Implementation

Three studies focused on identifying barriers [17,19] or facilitators [18] of CPG implementation. Five studies focused on measuring the clinicians’ awareness of, familiarity with, or attitude toward CPGs [16,22,23,25,26] in addition to exploring barriers and facilitators. Four studies examined clinicians’ compliance with certain guideline recommendations [14,19,20,24] and reasons for non-compliance. However, five studies (33%) did not focus on the investigation of barriers to and facilitators of guideline implementation. The first study used statistical analysis to identify the relationship between adherence to the guidelines and organizational culture [21]. The second study asked respondents one question about reasons for the lack of agreement with guidelines [27], whereas two studies questioned participants about reasons for not adhering to particular guideline recommendations [23,28]. One study [15] explored the views of cardiologists about the use of guidelines in clinical practice, where several quotations related to reasons for not using the guidelines were presented.

The most commonly identified barriers to CPG implementation were environmental factors, such as lack of time, resources, incentives, training and dissemination plans, and unavailability of treatment or diagnostic tests (Table 4; Appendix A
Appendix A). A few studies reported on (a) awareness of healthcare professionals regarding the existence of guidelines or familiarity with recommendations, (b) guideline characteristics (trustworthiness, clarity, and degree of complexity), and (c) patient factors. Lack of agreement with guidelines and preferences in clinical judgment, physician self-efficacy, and motivation were reported to a lesser extent.

Three studies discussed the facilitators of CPG implementation. Interviews and focus groups with decision-makers and experts in Iran [18] identified five categories for intervention to implement CPGs: health professional-oriented interventions, such as education, disseminations, and feedback, imposing financial incentives and payment mechanism, organizational intervention as good information infrastructure and staffing, regulatory interventions to stimulate development and implementation of guidelines, and multifaceted interventions. A focus group of 25 physicians identified several facilitators of CPG implementation: electronic medical records, easy access to computers within offices, organizational endorsement and quality monitoring, and a central committee who will supervise and conduct ongoing training [16]. A study involving 124 primary health care and family specialists used stepwise multiple linear regression analysis to determine the predictors of the implementation of smoking cessation counseling and therapy [25]. The results indicate that awareness of the guidelines for smoking cessation therapy, physician’s smoking status, being highly confident in the ability to provide smoking cessation counseling and therapy, reporting the ineffectiveness of smoking cessation therapy as a barrier, and perceived benefit of reducing patients’ physical symptoms were independently statistically significant predictors of the use of the guidelines.

## 4. Discussion

The systematic review retrieved a total of 15 studies assessing the barriers to and facilitators of CPG implementation. The most commonly reported barriers were environmental factors and lack of awareness of the existence of guidelines. Interestingly, all factors reported in Table 4 of barriers or facilitators in the MENA region were identified by a recently published meta review of 25 systematic reviews of qualitative, quantitative, or mixed-methods studies that identified barriers or facilitators for the implementation of CPG s form different countries [8].

The barriers identified in the current review are similar to those identified by previous research, such as lack of awareness of or familiarity with the existence of guidelines, lack of agreement with guideline recommendations, and limited work time and resources [3,6,7,8]. The facilitators identified include disseminating and advertising guideline materials, educating and training individuals about the guidelines, regulatory and financial incentives, and support from institutions, which are in agreement with findings from previous studies [3,7,8].

A review of studies from 2004 to 2013 that examined the dissemination and implementation strategies of guidelines on arthritis, diabetes, colorectal cancer, and heart failure identified 32 studies [4]. The majority of studies reported on strategies targeting professional approaches, such as educating and reminding groups about guideline intent and benefits and providing print materials, such as summaries, algorithms, or referral forms. Few studies used financial or organizational approaches, whereas no study used regulatory approaches. This finding was also observed in other reviews [29]. Thus, future studies should investigate the unused approaches, which may be effective for improving CPG dissemination and implementation.

Five of the identified studies, the guidelines examined were from international sources, whereas the remaining studies did not explicitly indicate the source or examined guidelines in general. The utility, applicability, and relevance of international recommendations for local settings require significant efforts from local experts [30]. This aspect is particularly important as our finding suggests that environmental factors, specifically lack of resources and availability of medications or diagnostic tests, and guideline factors, such as trustworthiness, are the most commonly reported barriers. Moreover, the quality, quantity, and spectrum of guidelines in low–middle income countries do not measure up to high-income countries [30,31,32]. Therefore, local experts need to make clear decisions about using existing evidence sources, adopting (with or without contextualizing), or adapting on the basis of several factors, such as scarce resource allocation [30].

The participants were mainly physicians. In other words, the perspectives of other healthcare practitioners on barriers to the utilization of CPGs are less understood. This point was observed in similar reviews [29]. Patient-centered care requires multidisciplinary expertise. As such, future research on barriers to CPG implementation should acknowledge this aspect.

### 4.1. Implications for Practice

This review is the starting point of future research in the MENA region for designing strategies that will improve compliance with CPGs in line with the identified barriers. Our findings indicate that a variety of barriers must be addressed to enhance adherence to CPG. Therefore, focusing on one barrier will result in less than the desired adherence to guidelines. Educational strategies, feedback on guideline compliance, providing reminders, institutional support, and investment in resources are examples of interventions needed for well CPGs implementation [33,34,35]. Using multifaceted strategies that target patients, clinicians, organizations, and policies is recommended [36]. Individuals responsible for designing implementation interventions and quality improvement projects should prioritize barriers and facilitators and design implementation strategies in a systematic manner. For instance, the checklist developed Flottorp et al. [37] can be used, which includes 57 potential determinants of practice (i.e., barriers and facilitators) grouped into seven domains, namely, guideline factors, individual health professional factors, patient factors, professional interactions, incentives and resources, capacity for organizational change, and social, political, and legal factors.

### 4.2. Implications for Research

The economic status of the 21 MENA countries is variable. Six countries are classified as high income (Bahrain, Saudi Arabia, Kuwait, Oman, Qatar, and United Arab Emirates), three as low income (Sudan, Syria, and Yemen), five as low–middle income (Egypt, Algeria, Morocco, Tunisia, West Bank, and Gaza) and the remainder consists of upper-middle-income countries. However, many countries are under-represented in the identified studies. Thus, the need emerges for further research on barriers to and facilitators of CPG implementation from the contexts of such countries.

The current review suggests that studies on barriers and facilitators in the MENA region are limited in number and quality. Especially, the limited number of studies using qualitative or mixed research methodology suggests a need for this type of research in the future. Qualitative methods are valuable in implementation research because they address what is happening in the implementation process and why [38]. Cross-sectional questionnaire-based research in the future should include representative samples with adequate sample sizes.

An interesting finding is that the majority of studies examined barriers to compliance with existing guidelines rather than barriers to implementing new guidelines. Our results indicate that the lack of protocols and processes for dissemination and implementation is a commonly reported barrier. Hence, future studies on the process of dissemination and implementation of new guidelines are warranted.

### 4.3. Strengths and Limitation

Despite the similarity of MENA countries in terms of ethnicity and religion, the review aggregated the barriers and facilitators from different healthcare systems, population size, and economic status [39]. This initiative ultimately led to the exploration of a wide array of factors that may remain unidentified if the research was specific to one country. The quality of identified studies was critically appraised. Moreover, the barriers and facilitators were categorized according to the known framework of barriers to adherence to guidelines [6] which improved the subjectivity of data extracted.

A major limitation of the study is that the search strategy excluded gray literature, and non-English language studies, thus introducing publication and language bias. Furthermore, poorly indexed studies and those not indexed in PubMed, EMBASE, and EBSCO databases may have been missed. Finally, the included primary studies have several weaknesses in their design and reporting, which limit the conclusion of this systematic review.

## 5. Conclusions

Studies on barriers and facilitators for compliance with CPGs in the MENA region are limited in number and quality. Evidence suggests that clinicians perceive environmental factors, specifically lack of time, resources, incentives, training and dissemination plans, and unavailability of treatment or diagnostic tests, as the main reasons for failure to implement CPGs.

## Figures and Tables

**Figure 1 healthcare-08-00564-f001:**
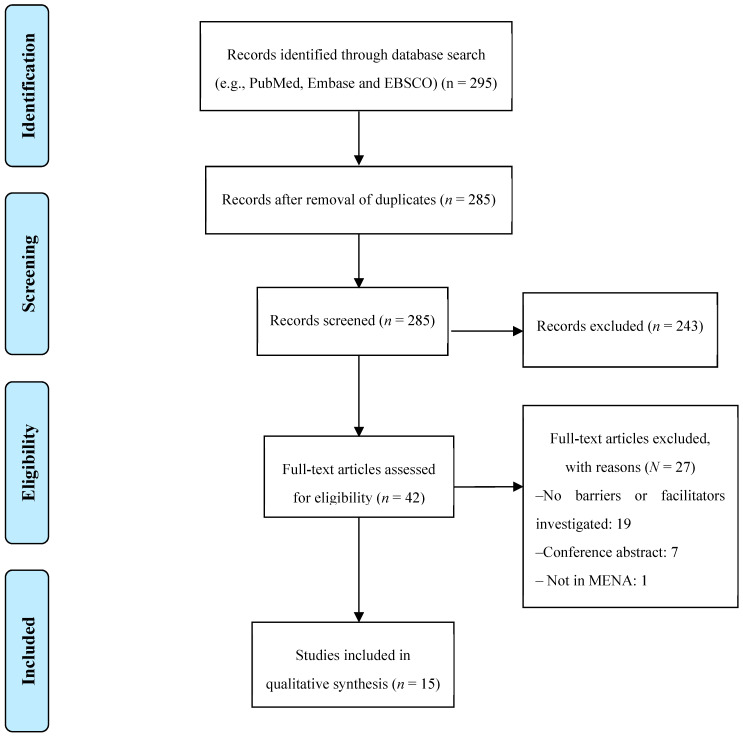
PRISMA flowchart for study selection.

**Table 1 healthcare-08-00564-t001:** Study Characteristics.

First Author, Year, Country	Study Objectives	Study Methods	Sample Size	Type of Participants (Setting)
Qualitative Studies				
Baradaran-Seyed Z., 2013 Iran	To identify the barriers and implementation of CPGs in Iran.	Interviews and 1 focus group	12 and 11	Health policy- and decision- makers, the experts of CPGs development, and the experts of EBM (General)
Majdzadeh R., 2013 Iran	To identify the strategies for application of CPGs produced in Iran.	Interviews and 1 focus group	12 and 11	Health policy- and decision-makers, the experts of CPGs development, and the experts of EBM (General)
Al Ketbi L.M., 2018 United Arab Emirates	To determine the barriers and facilitators of CPG implementation and to determine ways to improve the implementation of CPG recommendations	6 focus groups	25	Physicians (Ambulatory)
Radwan M., 2018, Palestine (West Bank and Gaza)	To assess the attitudes of Palestinian health care professionals toward the most perceived factors influencing the adherence to the CPG for diabetes mellitus	Interviews	20	Physicians + nurses who were managing/supervising the functions related to chronic diseases (Ambulatory)
**Mixed Methods**				
Wahabi H.A., 2012 Saudi Arabia	To examine the compliance of the healthcare providers in the Pediatrics Emergency Department, with the recommendations of the Pediatrics Asthma Management Protocol (PAMP), and to explore the reasons behind non-adherence.	Chart review and 2 focus groups and 4 interviews	10 and 10 and 4	Physicians + nurse + 1 pharmacist interviewed (Hospital)
Elsadig H., 2018, Sudan	To explore the views of prescribers in cardiology in Sudan about the use of guidelines in clinical practice and the extent to which guidelines whether national or international can be adopted in clinical practice in Sudan.	Interviews and survey	25 and 72	Physician (Hospital)
**Quantitative**				
Radwan M., 2017 Palestine (West Bank and Gaza)	Identifying the predominant culture within the Palestinian Primary healthcare centers settings and testing its role in the adherence to CPG for diabetes mellitus.	Questionnaire	323	Physicians and nurses (Ambulatory)
Radwan M., 2017 Palestine (West Bank and Gaza)	To explore adherence level and the most perceived barriers of the adherence to the CPG for diabetes mellitus	Questionnaire	323	Physicians and nurses (Ambulatory)
Alsubaiei M.E., 2017 Saudi Arabia	To assess assessed physicians’ knowledge of the Global Initiative for Chronic Obstructive Lung Disease (recommendations in terms of the diagnosis, assessment and management of patients with chronic obstructive pulmonary disease (COPD)	Questionnaire	44	Physicians (Hospital)
Salama A.A., 2010 Egypt	To assess the current situation as regard clinician attitude towards national and international guidelines for management of pediatric asthma and their adherence to its recommendations through written questionnaire.	Questionnaire	352	Physicians (Not clear)
Aloush SM., 2017 Jordan, Egypt, and Saudi Arabia	To evaluate nurses’ and hospitals’ compliance with ventilator-associated pneumonia prevention guidelines, the factors that affect their level of compliance, and barriers to compliance.	Questionnaire	471	Nurses (Hospital)
Thabet F.C., 2017 Saudi Arabia	To describe the initial management of pediatric patients with severe sepsis, to assess the compliance of this management with the 2006 the American College of Critical Care Medicine-Pediatric Advanced Life Support (ACCM-PALS) guidelines, and the 2012 surviving sepsis campaign (SSC) guidelines, and to identify barriers to adherence to these guidelines	Questionnaire	61	Pediatric intensivists (Hospital)
Sharif N.E., 2016 Palestine	To analyze the pattern of diabetes mellitus care by physicians and nurses and their self-reported compliance with the guidelines	Questionnaire	401	Physicians and nurses (Ambulatory)
Jradi H., 2015 Saudi Arabia	To assess knowledge, use, and barriers to the implementation of tobacco dependence (the 5A’s: Ask, Assist, Assess, Advise, and Arrange),	Questionnaire	124	Physicians (Ambulatory)
Al-Azzam S.I., 2012 Jordan	To assess the practice of surgical antibiotic prophylaxis and adherence of practitioners to the American Society of Health System Pharmacists (ASHP) guidelines for antimicrobial prophylaxis in surgery and to explore reasons for non-compliance.	Questionnaire	160	Surgeons (Hospital)

**Table 2 healthcare-08-00564-t002:** Quality assessment of qualitative studies.

First Author (Year)	1	2	3	4	5	6	7	8	9
Baradaran-Seyed Z. (2013)	Y	Y	Y	Y	C	C	C	C	Y
Majdzadeh R. (2013)	Y	Y	Y	Y	C	Y	Y	C	Y
Al Ketbi L.M. (2018)	Y	Y	Y	Y	Y	C	Y	Y	Y
Radwan M. (2018)	Y	Y	Y	Y	Y	C	Y	Y	Y
Wahabi H.A.* (2012)	Y	Y	Y	Y	Y	C	Y	Y	Y
Elsadig H. (2017) *	Y	Y	Y	Y	Y	C	C	C	Y

1. Was a clear statement on the aims of the research provided? 2. Was the qualitative methodology appropriate? 3. Was the research design appropriate for the aims of the research? 4. Was the recruitment strategy appropriate for the aims of the research? 5. Was the data collected in a manner that addressed the research issue? 6. Has the relationship between researcher and participants been adequately considered? 7. Have ethical issues been considered? 8. Was data analysis sufficiently rigorous? 9. Was a clear statement of the findings provided? Y: Yes, N: No, C: Can’t tell. * A mixed-methods study.

**Table 3 healthcare-08-00564-t003:** Quality assessment of quantitative studies.

First Author (Year)	1	2	3	4	5	6	7	8	9	10	11
Radwan M. (2017)	Y	Y	Y	N	Y	Y	Y	Y	Y	N	N
Radwan.M (2017)	Y	Y	Y	N	Y	Y	Y	Y	Y	N	N
Alsubaiei M.E. (2017)	Y	Y	Y	C	C	N	Y	C	Y	N	C
Salama A.A. (2010)	Y	Y	N	C	Y	Y	C	C	N	N	C
Aloush S.M. (2018)	Y	Y	N	Y	C	Y	Y	C	Y	N	C
Thabet F.C. (2017)	Y	Y	Y	N	Y	Y	Y	C	N	N	C
Sharif N.E. (2016)	Y	Y	Y	N	Y	Y	Y	C	Y	N	N
Jradi H. (2017)	Y	Y	N	C	C	N	N	C	Y	N	C
Al-Azzam S.I. (2012)	Y	Y	N	C	C	N	Y	C	N	N	C
Elsadig H. (2017)	Y	Y	C	C	C	N	Y	C	Y	N	C

1. Did the study address a clearly focused question/issue? 2. Is the research method (study design) appropriate for answering the research question? 3. Is the method of selection of the subjects (i.e., employees, teams, divisions, and organizations) clearly described? 4. Could the method for obtaining the sample introduce (selection) bias? 5. Was the sample of subjects representative of the population to which the findings will be referred? 6. Was the sample size based on pre-study considerations of statistical power? 7. Was a satisfactory response rate achieved? 8. Are the measurements (questionnaires) valid and reliable? 9. Was the statistical significance assessed? 10. Are confidence intervals given for the main results? 11. Were confounding factors overlooked? Y: Yes, N: No, C: Can’t tell.

**Table 4 healthcare-08-00564-t004:** Barriers to implementation of clinical practice guidelines.

Category	Barriers	Total Number of Mentions
Healthcare professional	Lack of awareness of the existence of guidelines	2
	Lack of familiarity with CPGs recommendation	3
	Disagreement with the recommendations of the CPG	3
	Doubts about the positive impact of CPGs on outcomes	2
	Preference for experience over CPGs	1
	Lack of effective communication, research, and self-learning skills	1
	Lack of healthcare professional motivation	3
Guidelines	Lack of clarity or complexity	4
	Outdated guidelines	1
	Guideline trustworthiness (i.e., evidence quality, content, and developer)	6
Patients	Language and literacy problems	1
	Lack of motivation, compliance, and knowledge to follow the recommendations	4
	Patient comorbidities, mobility problems, polypharmacy, and self-empowerment capacity	2
	Patients’ financial situation and occupational status	5
Environment	Lack of protocols and processes of dissemination and implementation	7
	Lack of resources (staff, equipment, and beds)	7
	Lack of time	2
	Lack of clinical audit and feedback	4
	Difficulties with availability of medicines or test	2
	Lack of financial incentives	3
	Lack of training	4
	Insurance does not cover recommendations	1
	Lack of regulation and supervision	1
	Lack of evidence-based culture and education	2
	Lack of policy makers supports	1
	Deficiencies in the referral of patients to services	1

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
