# Peer review of "Barriers to and Facilitators of Adherence to Clinical Practice Guidelines in the Middle East and North Africa Region: A Systematic Review"

_healthcare, 2020, doi:10.3390/healthcare8040564_

Round 1
Reviewer 1 Report
Paper by Doctor Almazrou SH et al. challenged barrier and facilitator of implementing clinical guidelines. I would like to provide comments to improve the construction of the paper of interesting topic.
[Major]
1. Methods: I consider that the way of measuring implementation of guidelines has not been scientifically established. I wonder how the researchers measured it. In the Methods section, I could not understand the measuring way.
2. I understand that this paper treated systematic review, and not meta-analysis. However, how did they consider that this study could be included, and that is not. In other words, in my opinion, it may difficult to evaluate barriers and facilitators of implementing clinical guidelines.
3. Is the title appropriate? As a reader, I wonder that 'compliance' or 'adherence' may be a key word when I read this paper.
4. Is there any similar research in the other countries? It would support limitation of this study.
Honestly speaking, I could not evaluate without bias this systematic review. It is because 1. selection criteria did not persuade me. 2. title. 3. measuring barrier and facilitator of implementation would be impossible. However, if the last question (no. 3) of mine has been solved, it would be welcome by academic society and clinicians. I hope that my comments would support the researchers.
Reviewer 2 Report
In this review Almazrou and colleagues review the barriers in implementing clinical practice guidelines in Middle East and North Africa by health professionals. The search strategy is fine. I have following suggestions to broaden the scope of the review.
1. In the supplementary table authors should include the dates they searched the terms for reproducibility and so that it can become clear if any new papers have been added.
2. In Table 4, include which barriers have been previously identified in other reviews and in other regions. This will help identify the universal and region specific barriers. Authors should discus in more details what factors are common and different these and other regions.
3. A guideline or suggestions to improve the implementation should be included and highlighted in the discussion section.
4. Minor typos or referencing issues such as line 121 should be fixed.
Round 2
Reviewer 1 Report
I consider that the researchers have addressed all of my comments. I have no more comment. I appreciate their efforts to report the important results.